# Effects of *Moringa oleifera* Leaf Powder on the Growth Performance, Meat Quality, Blood Parameters, and Cecal Bacteria of Broilers

**DOI:** 10.3390/vetsci11080374

**Published:** 2024-08-14

**Authors:** Md G. Akib, Al Rifat, Chondhon Bormon, Amitush Dutta, Mohammad Shamsul Ataher, Mahmoud Azzam, Mohammed Hamdy Farouk, Razib Das, Md Abul Kalam Azad, Shad Mahfuz

**Affiliations:** 1Department of Animal Nutrition, Faculty of Veterinary, Animal and Biomedical Sciences, Sylhet Agricultural University, Sylhet 3100, Bangladesh; akib.vabs@student.sau.ac.bd (M.G.A.); rifat.vabs@student.sau.ac.bd (A.R.); chondonb12@gmail.com (C.B.); amitushduttavet86@gmail.com (A.D.); shamsulataher1981@gmail.com (M.S.A.); 2Animal Production Department, College of Food and Agriculture Sciences, King Saud University, Riyadh 11451, Saudi Arabia; mazzam@ksu.edu.sa; 3Animal Production Department, Faculty of Agriculture, Al-Azhar University, Nasr City, Cairo 11884, Egypt; mhfarouk@azhar.edu.eg; 4Department of Human Nutrition, Food and Animal Sciences, College of Tropical Agriculture and Human Resources, University of Hawai’i at Manoa, Honolulu, HI 96822, USA; 5Institute of Subtropical Agricultural, Chinese Academy of Sciences, Changsha 410125, China; azadmak@isa.ac.cn

**Keywords:** *Moringa oleifera* leaf powder, broilers, growth performance, meat quality, blood parameters, cecal bacteria

## Abstract

**Simple Summary:**

This study evaluated the effects of *Moringa oleifera* leaf powder (MLP) on the growth performance, meat quality, carcass traits, blood profile, and cecal bacteria of broiler chickens. MLP-fed broilers showed the highest weight gain, average daily feed intake, and best feed efficiency, as evidenced by a lower feed conversion ratio. MLP also enhanced meat quality, indicated by higher pH, lower cooking loss, and reduced cholesterol and triglyceride levels. Positive impacts on the blood profile included increased red blood cell count, higher hemoglobin levels, and a lower stress indicator (H/L ratio). Additionally, MLP positively changed the cecal bacterial population by reducing harmful *E. coli* and *Shigella* spp. while increasing beneficial *Lactobacillus* spp. Thus, MLP is a promising natural feed additive for improving broiler growth, meat quality, overall health, and producing antibiotic-free and healthier broilers for consumers.

**Abstract:**

The effect of dietary inclusion of *Moringa oleifera* leaf powder (MLP) on the growth, meat quality, carcass characteristics, hematobiochemical profile, and cecal bacteria of broiler chicken was investigated in this research trial. In this study, 192-day-old Arbor Acre broiler chicks were assigned in a completely randomized design to three groups: control, antibiotic, and MLP. A standard basal diet was given to the control group, while the antibiotic group received 75 mg/kg chlortetracycline, and the MLP group received 100 mg/kg *M. oleifera* leaf powder supplemented basal diet. Each group was further divided into eight replicates consisting of eight birds each, and the trial ran for 35 days. Among the groups, the MLP-fed broilers achieved the highest final body weight (FBW), average daily gain (ADG), and average daily feed intake (ADFI). Notably, the FCR for the whole experimental period was lower in the MLP group, indicating a more efficient use of feed for growth. Supplementation of MLP with basal diet significantly increased (*p* < 0.05) the weight of thighs and drumsticks relative to live weight %, while the spleen and abdominal fat weight (% of live weight) were significantly decreased (*p* < 0.05). Adding MLP to the diet improved the meat quality of broilers, as indicated by the highest pH of meat at 45 min and the lowest cooking loss (%) observed in this group. MLP exhibited hypocholesterolemic and hypolipidemic effects, with the lowest total cholesterol and triglyceride levels compared to non-supplemented groups. The hematological profile revealed that the MLP group exhibited the highest RBC count and Hb level, while also showing the lowest H/L ratio. Moringa supplementation significantly (*p* < 0.05) modulated the cecal bacterial population, reducing pathogenic *E. coli* and *Shigella* spp. while increasing beneficial *Lactobacillus* spp. and the total aerobic plate count (TAPC). In conclusion, *Moringa oleifera* leaf powder (MLP) can be used as a natural feed supplement for promoting the growth, meat quality, healthy blood, and sound health of broilers.

## 1. Introduction

*Moringa oleifera*, a member of the Moringaceae family, is widely cultivated and cherished for its nutritional value. Originating in the Indian subcontinent, this fast-growing, drought-tolerant tree thrives in a diverse range of rainfall and soil conditions, rendering it prevalent in tropical regions and a valuable source of essential nutrients worldwide [1]. With many colloquial names like Moringa, Drumstick tree, Horseradish tree, Malunggay, Mother’s best friend, and Benz olive tree, this versatile plant is known as Sajna or Sajina in Bangladesh [2,3]. The Moringa plant is also called the ‘Miracle Tree’, as it is a true marvel that has been extensively studied for its potential medicinal applications in over 300 diseases [4]. Furthermore, different parts, from its roots to leaves, of this remarkably multipurpose tree are used as fodder, herbal remedies, aromatic spices, nutritious food, natural coagulants, bee-friendly nectar, water purifier, reliable fuel, and even fertilizer [5].

Moringa’s high nutritional value makes it a versatile food source, enjoyed by both humans and animals, including chickens [6]. Scientific studies show that *M. oleifera* leaves are a nutritional powerhouse, containing high-quality protein, energy sources, essential amino acids, calcium, iron, potassium, phosphorus, and a variety of vitamins (A, D, E) [1,7]. They are also rich in antioxidants like vitamin C, beta-carotene, phenols, and flavonoids. This impressive array of nutrients contributes significantly to overall health and well-being. Additionally, their low levels of tannins and inhibitors of enzymes like trypsin, lipase, and amylase make them easily digestible and allow for optimal nutrient absorption [8]. The nutrient composition of *M. oleifera* indicates its potential to serve as a micronutrient source and dietary supplement for poultry. Studies show that adding up to 5% of *M. oleifera* leaf in powdered form into the diet exerts a positive impact on broilers without any adverse effects by improving growth performance, blood profile, antioxidant status, meat quality, carcass characteristics, and the prevention of gut pathogens [6,9,10].

Previously, the poultry industry heavily relied on antibiotic growth promoters (AGPs) to enhance enteric health and disease control. However, due to concerns about antibiotic resistance and public health, European countries banned the use of in-feed antibiotics as growth promoters on 1 January 2006 [11], while Bangladesh implemented a similar ban on 28 January 2010 [12]. As a result, several alternative additives, including phytobiotics, organic acids, enzymes, bacteriophages, probiotics, and prebiotics, are being investigated as substitutes for AGPs [13]. *M. oleifera* emerges as a promising alternative to AGPs. ‘Pterygospermin’, a compound found in *M. oleifera* that readily dissociates into benzyl isothiocyanate, exhibits potent antibiotic and antifungal properties, suggesting its potential as a natural growth promoter for poultry [2]. Additionally, *M. oleifera* leaf extracts are able to inhibit the growth of human enteric pathogenic bacteria, including *Escherichia coli*, *Pseudomonas aeruginosa*, *Shigella* spp., *Staphylococcus aureus*, and *Streptococcus pyogenes*, at different concentrations [14]. Research suggests that *M. oleifera* extracts or compounds may offer protection against cancer, diabetes, inflammation, high blood pressure, and even liver damage [15,16,17]. *M. oleifera* leaves and root bark also boast antiseptic properties, which are reported to strengthen the body defense mechanism by hindering pathogen proliferation and mitigating the harmful endotoxins released by them [18,19]. *M. oleifera* also demonstrates hypolipidemic, hypocholesterolemic, and anti-atherosclerotic properties, indicating its potential therapeutic use in preventing cardiovascular diseases [20,21]. Alternatively, *M. oleifera* leaves, abundant in polyunsaturated fatty acids (PUFA), could indirectly benefit human health. Studies suggest that including *M. oleifera* leaf powder in chicken feed increases the PUFA content in their meat, making it potentially healthier for human consumption due to PUFA’s cardiovascular and immune system advantages [22,23].

Consumer concerns about the negative health impacts of chemical growth promoters in poultry feed are driving the demand for organic meat and eggs. This has led to a surge in interest in natural alternatives like *M. oleifera*. While research suggests that *M. oleifera* offers potential benefits, the optimal dosage for poultry diets and its exact mechanism of action remain unclear [9]. Additionally, *M. oleifera* contains anti-nutritional properties such as saponins, tannins, phytates, and cyanogenic glycosides [24]. The uncontrolled use of *M. oleifera* in broiler diets could have detrimental effects. Therefore, to address this knowledge gap, this study investigated the effects of small concentrations of MLP on the overall growth performance, carcass characteristics, meat quality, hematobiochemical profile, and cecal bacteria in broiler chickens.

## 2. Materials and Methods

### 2.1. Experimental Design and Dietary Treatments

In this study, 192-day-old commercial broiler chicks (Arbor Acres, AA) of either sex were weighed (initial body weight, 40.03 ± 0.69 g) and assigned completely randomly into three treatment groups with 8 replications and 8 chicks per replication (64 birds per treatment), namely control, antibiotic, and *Moringa oleifera* leaf powder (MLP). The ration formulation for broilers was conducted in accordance with the National Research Council, 1994 [25] guidelines, comprising a starter feed for the period from 0 to 21 days and a finisher feed for the period from 22 to 35 days, both designed to meet dietary nutrient requirements. The control group was given a basal diet, whereas the antibiotic group received the same basal diet with an addition of 75 mg/kg of Chlortetracycline, and the MLP group was fed a basal diet supplemented with 100 mg/kg of *M. oleifera* leaf powder. Table 1 shows all of the experimental feed ingredients and nutrient compositions used in this trial.

### 2.2. Study Site and Managements

This study, conducted at the broiler research shed of the Department of Animal Nutrition at Sylhet Agricultural University (Sylhet-3100), was authorized by the university’s Animal Care and Use Committee. The research shed was an open-system poultry house located in an east–west direction to avoid direct sunlight. The temperature and relative humidity (RH) were controlled by a proper lighting system, curtain management, and natural ventilation. The initial temperature of the brooding house was 37 °C, and it was gradually decreased at a rate of 3 °C per week until reaching the end of the experiment. The initial RH within the poultry shed was 55% and was progressively raised by 5% each week, reaching a final level of 70%. Broilers were reared on a deep litter floor and were given an adequate quantity of experimental diets and ad libitum clean water daily. The birds were immunized by a standard vaccination schedule according to conventional immunization procedures. All management procedures were performed according to the recommendations for raising AA broilers [26].

### 2.3. Preparation and Chemical Analysis of M. oleifera Leaf Powder

*M. oleifera* leaves of different mature stages were collected from the local area, then mixed uniformly and cleaned properly. Collected leaves were spread under shade with constant turning for 3 days to decrease moisture thereafter; these were dried at a constant temperature of 30 °C for 30 min in an oven to make them crispy for easy blending. Later, leaves were ground up to powder using a blender machine.

The MLP samples in triplicate were analyzed following the guidelines outline by the Association of Official Analytical Chemists [27] for dry matter (DM), ether extract (EE), crude protein (CP), crude fiber (CF), total minerals (Ash), amino acids (AA), calcium, and phosphorus. On the whole, the Kjeldahl method was followed to estimate protein content. Fat content was extracted from petroleum ether samples in a Soxhlet apparatus. For the determination of Ash content, samples were incinerated in a furnace at 600 °C for 6 h. Moisture content was obtained by subtracting dry matter from 100. Organic matter (OM) was calculated by subtracting the estimated ash value% and moisture% from 100 [3].
OM = {100 − (Moisture% + Ash%)}(1)

Nitrogen free extract (NFE) was calculated by the following equation [28]:NFE = 100 − (Moisture% + Crude protein% + Crude fiber% + Crude fat% + Ash%)(2)

The analysis of amino acids was performed with an automated amino acid analyzer (L-8900, Hitachi, Tokyo, Japan). To prepare the samples, they were first hydrolyzed using 6 N hydrochloric acid (HCl) at 110 °C for 24 h under vacuum [29].

To determine the phosphorus (P) and calcium (Ca) content of MLP, a protocol obtained from Sultana [3] was followed. The protocol involved digesting the samples first with a solution of nitric acid, perchloric acid, and double-distilled water. Then, using that, the calcium content was measured by complexometric titration, and the phosphorus content was determined colorimetrically. The analyzed chemical and amino acid composition of MLP are presented in Table 2 and Table 3, respectively.

### 2.4. Growth Performances of Broilers

Every bird in each replication was weighed using a Miyako ACS-A9 digital weight scale (Miyako Appliance Limited, Dhaka, Bangladesh), and the average weight for each replication was calculated. The growth performances of broilers were observed and measured by calculating average daily gain (ADG), average daily feed intake (ADFI), and feed conversion ratio (FCR). For this purpose, the following formulas were used:ADFI = {(Feed offered to the broilers per replicate; gm) − (residual feed on the feeder; gm + feed waste; gm)}/ (total number of broilers per replication × total number of days)(3)
ADG **=** {(Total weight measured per replicate before a week; gm) − (Total weight measured per replication after a week; gm)}/(total number of broilers × total number of days)(4)
(5)$$FCR=\frac{Average\;daily\;feed\;intake\;(ADFI)}{Average\;daily\;gain\;(ADG)}$$

### 2.5. Measurements of Carcass Characteristics and Inner Organs Weight (%)

At 35 days of age, one broiler from each replicate (8 from each diet group) was randomly selected and slaughtered by cervical disarticulation for evaluation of carcass traits, inner organs weight, and required sample collection. The dressed weight was obtained after removing the head, skin, feathers, shanks, intestines, and offal, including giblets from the carcass. Thighs, drumsticks, wings, shanks, and internal organs, along with abdominal fat, were excised from all slaughtered broilers and then weighed. The internal organs include the heart, liver, proventriculus, gizzard, bursa, spleen, pancreas, kidneys, and intestines. Each bird was weighed immediately before being sacrificed, and the relative organ weights were presented as a percentage of its live weight.

### 2.6. Meat Quality Analysis

#### 2.6.1. pH Value of Breast Muscle

After slaughtering, the pH values of the breast muscle were measured using a probe equipped with an electrode from a portable pH meter (Orion model 301; Orion, Beverly, MA, USA) at 45 min and 24 h. Prior to measurement, the probe was calibrated using pH 4 and pH 7 calibration buffers. Readings were taken from at least three locations of each sample, and the average values were considered the final value.

#### 2.6.2. Drip Loss Determination

To determine the drip loss of meat, a method described by Cheng et al. [30] was followed accordingly. Initially, required meat samples were trimmed, weighed, and placed in inflated plastic bags. These bags were then hung for 24 h and 7 days at 4 °C. After each hanging period (24 h and 7 days), the samples were reweighed after wiping the surface with filter paper. The percentage of drip loss was calculated based on the weight difference before and after hanging. Drip loss was calculated by using the following equation:(6)$$Drip\;loss\%=\frac{Intial\;weight\;of\;fresh\;meat\;-\;weight\;after\;thawing}{Initial\;weight\;of\;fresh\;meat}\times 100$$

#### 2.6.3. Cooking Loss

To determine the cooking loss, a 100 g sample of breast meat was sliced and sealed in a plastic bag. The sample was then cooked in a preheated water bath until the internal temperature reached 75 °C for 5 min. A digital needle-tipped thermometer (H 1145, Hanna Instruments, Padova, Italy) was used to monitor the internal temperature of the meat. After cooking, the samples were removed from the water bath, and water was drained from the bag. Then, the samples were promptly chilled in running water until they reached room temperature. Excess surface moisture was removed with paper towels before final weighing of the cooked sample. The cooking loss was calculated by using the equation below [31]:(7)$$Cooking\;loss\;\%=\frac{weight\;of\;fresh\;meat\;-\;weight\;of\;cooked\;meat}{weight\;of\;fresh\;meat}\times 100$$

### 2.7. Blood Sample Preparation for the Analysis of Hemato-Biochemical Parameters of Blood

Blood samples were collected in the evening at day 35 for hematobiochemical analysis. From each replicate, one broiler was randomly chosen (8 broilers from each group), and then two aliquots of blood samples were collected from their wing vein. The first sample was taken in a tube containing EDTA (anticoagulant) for the determination of various hematological parameters, including red blood cell (RBC) count, hemoglobin (Hb) level, packed cell volume (PCV), mean corpuscular volume (MCV), mean corpuscular hemoglobin concentration (MCHC), red cell distribution width–coefficient of variation (RDW–CV), total leukocyte counts (WBCs), and differential leukocyte counts (DLCs), following standard hematological procedures by using an automated H 360 hematology analyzer (Erba Mannheim, Mannheim, Germany).

On the other hand, the second blood aliquot was collected without an anticoagulant and allowed to clot. The resulting serum was then separated and stored at −20 °C for later analysis. Blood biochemical properties (creatinine, total cholesterol, serum glutamic pyruvic transaminase, triglyceride) were measured by using a Dade Behring Dimension RXL Max Chemistry Analyzer (Manufacturer: Siemens Healthineers, Erlangen, Germany).

### 2.8. Cecal Bacteria Count

One gram of cecal content from a randomly chosen broiler per replicate group was collected and placed into Eppendorf tubes filled with phosphate-buffered saline (PBS) for preservation. Then, using sterile saline, a series of tenfold dilution up to 10^−5^ was prepared in the laboratory. Selective agar media were used for culturing specific bacterial species employing conventional microbiological techniques. For the *Escherichia coli* and *Klebsiella* spp. Culture, two concentrations (10^−4^ and 10^−5^) were selected. A 20 μL portion of the dilution was carefully absorbed by a micropipette, evenly spread onto a MacConkey Agar medium, and then incubated for 24 h at 37 °C in a constant-temperature incubator. On the MacConkey agar, *E. coli* formed pink to red, small, round, smooth, and slightly raised colonies, while *Klebsiella* spp. formed dark pink, mucoid, larger, and more prominent colonies [32]. The resulting colonies were then enumerated using a colony counter. The following formula was used for colony counting:(8)$$CFU/gm=\frac{Average\;number\;of\;colonies\;\times \;Dilution\;factor\;}{Mass\;of\;culture\;sample\;platted\;(gm)}$$

In our research, the dilution factor was 10^4^ and 10^5^, as it is the reciprocal of the dilution ratio, and the mass of the culture was 0.02 gm.

The same method was used for culturing the remaining specific cecal bacteria species. *Lactobacillus* spp. was cultured on a *Lactobacillus* MRS agar (LMRS), resulting in the appearance of rounded, medium-sized, cream-colored colonies. *Salmonella* spp. and *Shigella* spp. were cultured on an SS (*Salmonella–Shigella*) agar, which resulted in *Salmonella* spp. forming translucent colonies with black centers, while *Shigella* spp. produced colorless colonies [33]. A nutrient agar (Manufacturer: HiMedia Laboratories LLC, Mumbai, India) was used as the culture medium for the total aerobic plate count (TAPC), resulting in the formation of colonies with irregular morphology. The results were calculated as the logarithm base 10 of colony-forming units (CFUs) per gram of cecal content. Each sample was replicated twice.

### 2.9. Statistical Analysis

Each replicate served as the experimental unit. Data were collected and analyzed using Excel 2023 (Microsoft Corp., Redmond, WA, USA) and SPSS version 27.0 (IBM Corp., Armonk, NY, USA), respectively. Intergroup differences were evaluated using one-way ANOVA, followed by Duncan’s multiple range test to compare treatment means. Statistical significance was defined at less than 0.05 (*p* < 0.05).

### 2.10. Ethical Statement

This experiment was conducted following the principles and guidelines outlined in the third edition (2010) of the Guide for the Care and Use of Agricultural Animals in Research and Teaching. The study was approved and supervised by the Animal Care and Use Committee at Sylhet Agricultural University, Sylhet, Bangladesh (No. AUP2023025).

## 3. Results

### 3.1. Growth Performances of Broiler

The effects of feeding *M. oleifera* leaf powder (MLP, at 100 mg/kg feed) on the growth performance of the broilers compared to the control and antibiotic (CTC, at 75 mg/kg feed)-fed groups are shown in Table 4.

The final body weight (FBW) was significantly higher (*p* < 0.05) in the MLP group compared to the control and antibiotic-fed groups. The average daily gain (ADG) was significantly higher (*p* < 0.05) in the MLP-fed group than the control and antibiotic-fed groups on days 8–14, 15–21, 22–28, and 29–35, including the whole experimental period (1–35 days).

The average daily feed intake (ADFI, g/d) was significantly higher (*p* < 0.05) in the MLP-fed group in comparison with the control and antibiotic-fed groups in the whole experimental period (1–35 day). On a weekly basis, the ADFI (g/d) was significantly higher (*p* < 0.05) in the MLP-fed group during the periods of days 8–14, 15–21, and 29–35 than that of the control and antibiotic groups. However, the feed intake was highest in the control group on days 22–28. There was a significantly lower (*p* < 0.05) FCR in the MLP-fed group than the other two groups on days 22–28 and the overall experimental period (1–35 days). However, the FCR was not affected on days 1–7, 8–14, 15–21, and 29–35 of the evaluation.

### 3.2. Carcass Characteristics

The drum stick weight (%) was significantly higher (*p* < 0.05) in the MLP group than in the control and antibiotic-fed groups, and the thigh wt.% was higher in the MLP group compared to the control group (Table 5). However, there were no significant differences in the dressed wt.%, wing wt.%, and shank wt.% of the broilers among the groups (Table 5).

### 3.3. Inner Organs Weight

The data from Table 6. show that feeding MLP to broilers significantly reduced (*p* < 0.05) spleen weight% and abdominal fat% (% live wt.) compared to control and antibiotic-fed groups. However, no significant changes were observed in the liver, pancreas, proventriculus, gizzard, intestine, and bursa among the treatments.

### 3.4. Meat Quality

A significantly higher (*p* < 0.05) pH (45 min) was observed in both MLP and antibiotic-fed groups than the control group. However, no significant differences in pH (24 h) were noted in this trial (Table 7). Cooking loss (%) was the lowest (*p* < 0.05) in the MLP-fed group compared to the remaining groups. However, no significant differences were observed in the drip loss (%) at 24 h and 7 days among the treatments (Table 7).

### 3.5. Biochemical Parameters of Blood

The dietary effect of MLP on the biochemical parameters of the blood of broilers compared with the control and antibiotic-fed groups is presented in Table 8. A significantly lower (*p* < 0.05) amount of total cholesterol and triglycerides (TG) were found in the MLP-fed group compared to the control and antibiotic-fed groups. However, the blood creatinine and SGPT levels did not show any significant differences (*p* > 0.05) among the experimental groups.

### 3.6. Hematological Parameters

The data in Table 9 represent the dietary effects of MLP-feeding on the hematological parameters of broilers in comparison with the control and antibiotic-fed groups of broilers.

The hemoglobin and RBC concentrations were significantly higher (*p* < 0.05) in the MLP-fed broilers than in the control and antibiotic-fed groups. In the case of WBC–DLC, H/L was significantly lower (*p* < 0.05) in the MLP-fed group than the other two groups. However, there was no significant effect of MLP feeding on PCV%, MCV, MCH, MCHC, RDW–CV%, WBC (total count), WBC–DLC (heterophil%, neutrophil%, monocyte%), and platelet (total count) of broilers.

### 3.7. Cecal Bacteria

The effect of MLP-feeding on the cecal bacteria of broilers is shown in Table 10. *Escherichia coli* had a significantly lower (*p* < 0.05) count in the MLP and antibiotic-fed groups than in the control group. A significantly lower (*p* < 0.05) count of *Shigella* spp. was observed in the MLP-fed group than in the others. On the other hand, a significantly higher (*p* < 0.05) count of *Lactobacillus* spp. was found in MLP-fed broilers compared to the remaining groups. The total aerobic plate count was significantly higher (*p* < 0.05) in both the MLP and control groups in comparison with the antibiotic group.

## 4. Discussion

This study demonstrates the potential effects of dietary supplementation *Moringa oleifera* leaf powder (MLP) on broiler production parameters, including growth performance, meat quality, carcass characteristics, hematobiochemical properties, and cecal bacteria. *M. oleifera* leaves are reported to be a good source of essential fiber, amino acids, vitamins (beta carotene, B-complex, C, K), essential minerals, and highly digestible proteins [3,9]. The leaves are also rich in highly potent hepatoprotective and anti-inflammatory properties, antibiotic (benzyl isothiocyanate) properties, higher amounts of polyphenols, flavonoids (kaempferol, quercetin), and phenolic acids [2,34,35].

The average daily feed intake of MLP-fed broilers was highest in the whole experimental period, which is in line with the research findings of El Tazi [10], who reported increased feed intake with MLP up to 5%. But, this finding contradicts Onunkwo and George [36], who observed reduced intake with MLP supplementation. We propose that active phytochemicals in MLP, like flavanol glycosides and moriginine, might explain the difference. These compounds, as suggested by Jamroz et al. [37] and Mbikay [7], could stimulate appetite, regulate blood sugar (glucose homeostasis), and enhance digestive enzyme production, leading to higher feed intake in broilers fed MLP. Consistent with higher feed intake, broilers fed MLP diets exhibited the highest final body weight (FBW) and average daily gain (ADG). This aligns with Abu Hafsa et al. [20] and Alwaleed et al. [38], who reported similar findings at various MLP inclusion levels (0.5%, 1%, 3%). The higher ADG in our study might be attributed to antibiotic properties, high polyphenols, flavonoids, highly digestible protein and amino acid contents, and low tannin content in the diet, which ultimately gives a higher body weight. Feed utilization also improved in the MLP-fed group, as they showed the lowest FCR in the fourth week and the overall experimental period, which is in accordance with Agashe et al. [39]. Collectively, synergy among the beneficial bioactive plant chemicals of *M. oleifera* exerts a positive effect on the digestive physiology of broilers, which results in a better digestion and utilization of feed [40]. On the contrary, a higher dose of MLP, above 5%, resulted in certain adverse effects on broilers, like reduced ADG, ADFI, FCR, and FBW, due to the high content of naturally occurring phytochemicals such as phenols, alkaloids, phytates, and tannins [41,42].

The thigh and drumstick wt. (%live weight) were significantly higher in the MLP-fed group compared to the control group, which is confirmed by the research findings of Makanjuola et al. [43] and El Tazi [10], who also found that the drum stick wt. % was significantly higher in the *M. oleifera* leaf meal (MOLM: 0.4%, 0.6%, 3%, 5%, 7%)-fed group than the control group.

There was a significant reduction (*p* < 0.05) in spleen weight (%live weight) in the MLP-fed broilers, which is supported by the research findings of Makanjuola et al. [43], who also found a reduction in spleen weight in broilers fed an MOLM-supplemented diet. The reduction in spleen weight may be attributed to immunomodulatory properties such as isothiocyanates and flavonoids, which decrease the workload of the spleen by reducing inflammatory reactions and enhancing specific immune responses [34,44]. Additionally, *M. oleifera* leaves are an abundant source of certain vitamins and essential minerals, in combination with other dietary elements, and could influence spleen function or size, but further investigation is needed to elucidate this potential effect [9,45]. In contrast to the findings of this study, Teteh et al. [46] found that adding 1–2% of *M. oleifera* leaves to the diet increased the spleen’s relative organ weight. The MLP-fed broilers exhibited significantly lower abdominal fat deposition compared to the other groups, which is in accordance with the research findings of Abu Hafsa et al. [20], who reported a reduced abdominal fat percentage in broilers fed diets containing varying levels (0.5%, 1%, 5%) of MLP. *M. oleifera* leaves could reduce fat deposition by three possible mechanisms. First, it might suppress gene expression involved in fat creation, leading to lower lipid biosynthesis [47]. Second, increased proliferation of beneficial gut bacteria like *Lactobacillus* spp. (mediated by MLP) could indirectly decrease fat synthesis by reducing the activity of the rate-limiting enzyme, acetyl-CoA carboxylase [20]. Finally, the presence of polyunsaturated fatty acids (PUFA) in the leaves might trigger faster lipid burning through FA β-oxidation, ultimately resulting in lower fat deposition [22].

Our study revealed a positive effect of MLP supplementation on broiler meat quality. The breast muscle from MLP-fed broilers exhibited a significantly higher initial pH (45 min post-mortem) compared to the control group. This aligns with Rehman et al. [48] who observed a higher pectoral muscle pH in the MLP-supplemented group. *M. oleifera* leaves are rich in certain alkaline minerals, such as calcium, magnesium, and potassium [3], as well as alkaloid chemicals like moriginine [7]. Based on this composition, we can hypothesize that supplementing MLP may increase the pH of meat by acting as an alkaline buffer that neutralizes acid byproducts. The cooking loss (%) was lowest in the MLP-fed group compared to the non-supplemented groups, supporting the findings of Azeezah et al. [49], who reported lower cooking loss in cockerel and egg-type chickens fed diets with MOLM at 0.5% and 1.0% concentrations. The richness of vitamin E and selenium in MLP potentially aids in stabilizing muscle membranes through antioxidant activation, preventing free radicals and reducing water loss from muscle cells [50,51]. Additionally, the higher initial muscle pH in the MLP-fed groups may contribute through better preservation of myofibril integrity and reducing protein denaturation [48]. Finally, phenolics and flavonoids present in MLP could interact with muscle proteins, enhancing cross-linking, preventing muscle oxidation and improving moisture retention during cooking [52,53]. These multi-pronged effects of MLP likely combine to reduce cooking loss in broiler meat.

The plasma levels of total cholesterol (TC) and triglycerides (TG) were lowest in the broilers treated with MLP compared to both the control and antibiotic-fed groups. Our findings are consistent with previous research by Abu Hafsa et al. [20], Ajantha et al. [54], and Dey and De [55], all of whom reported reduced TC and TG levels in broilers supplemented with *M. oleifera* leaves (0.075% to 5%). This significant reduction in plasma TC and TG levels observed in broilers can likely be attributed to its high content of certain phytochemicals, such as polyphenols, flavonoids, sterols, alkaloids, and phenolic compounds found in MLP [54]. Polyphenols found in *M. oleifera* leaves hinder pancreatic lipase activity, impeding triacylglycerol absorption. They also suppress β-apoprotein synthesis in intestinal mucosa, cholesterol esterification, and intestinal lipoprotein production [21]. Additionally, *M. oleifera* leaves contain sitosterol, which might decrease cholesterol levels by promoting fecal excretion (in the form of steroid) and reducing endogenous cholesterol absorption [17]. Another possible explanation is that *M. oleifera* activates lipoprotein lipase, which promotes cholesterol uptake from blood vessels, and augments lecithin cholesterol acyltransferase activity in HDL, resulting in reduced cholesterol levels through improved scavenging and excretion [21].

Our study demonstrated increased RBC count and Hb levels in broilers fed MLP diets, which is in line with Abbas et al. [56], who observed similar elevations in broilers supplemented with 0.75% and 1% MOLM. This consistency suggests a positive effect of MLP supplementation on broiler health. The observed increase in RBC count and Hb levels may be attributed to the blood-boosting properties of *M. oleifera,* characterized by its rich content of high-quality protein and diverse array of essential amino acids, which are the building blocks for hemoglobin production within red blood cells [57]. Additionally, the iron (Fe) content in *M. oleifera* plays a vital role in hemoglobin and myoglobin formation, directly influencing red blood cell (RBC) count and hemoglobin (Hb) value, critical for the transport and storage of oxygen [58]. Elevated RBC count and Hb levels indicate an enhanced capacity for oxygen transport and waste removal, potentially contributing to a healthier overall state in broilers [59]. The H/L ratio (characterized by decreased heterophils and increased lymphocytes) significantly decreased in broilers treated with MLP compared to both the control and antibiotic-treated groups, aligning with the findings of Hassan et al. [60] and Meel et al. [61], who independently reported a significant decrease in the H/L ratio with MOLM supplementation ranging from 0.1% to 2.0% in the basal diet, corroborating our results. According to Gross and Siegel [62], further supported by Hassan et al. [60], the H/L ratio serves as a dependable marker of long-term stress, with stressed birds typically displaying elevated H/L ratios. Therefore, the decreased H/L ratio suggest reduced stress in the MLP-treated birds compared to the non-supplemented groups. Additionally, a decreased H/L ratio correlates with enhanced immune responses [60]. The antioxidant and antimicrobial properties of *M. oleifera* likely contributed to the decrease in the H/L ratio, indicating an enhanced ability to combat infection. However, our findings contrast with those of Abu Hafsa et al. [20], who reported that supplementing *M. oleifera* leaves (MOL) could significantly increase the H/L ratio in broilers compared to control groups, which contradicts our result.

The gut microbiome plays a crucial role in the nutrition and health of broilers by aiding in digestion, especially with dietary fibers, and contributing to the synthesis of beneficial substances like B vitamins and amino acids [63]. In our study, we observed significantly lower numbers of *E. coli* and *Shigella* spp., but a higher count of *Lactobacillus* spp. in the ceca of broilers supplemented with MLP, compared to the control group. Similarly, Abu Hafsa et al. [20], observed a decreased colony count of *E. coli* but an increased count of *Lactobacillus* spp. in broilers supplemented with MOL at concentrations of 0.5%, 1%, and 5% with basal diet compared to the control group. In a separate study, Sharma et al. [64] also observed a reduction in coliform counts across all levels of the MOL-supplemented group compared to the control and even the antibiotic-treated groups. This consistency across studies strengthens the evidence that MLP supplementation not only reduces the levels of harmful microorganisms but also promotes the growth of beneficial bacteria among broilers. There was a significantly higher colony count of TAPC in the control and MLP-treated groups of broilers compared to antibiotic-fed broilers. The reduction in harmful bacteria might be attributed to a short polypeptide named 4 (V-L-rhamnosyloxy) benzyl isothiocyanate found in *M. oleifera*. This compound may act directly on microorganisms, disrupting the synthesis of cell membranes or essential enzymes, thereby inhibiting their growth [65]. MLP could modulate the metabolic function and intestinal microbiota, resulting in the higher production of cecal SCFA [6]. In addition, *M. oleifera* leaf powder could activate the epithelial and immune cells that protect the intestinal barrier and maintaining gastrointestinal homeostasis in broilers [6,63]. This regulates microbial diversity and increases the abundance of beneficial cecal bacteria.

## 5. Conclusions

The findings of this study demonstrate that *Moringa oleifera* leaf powder (MLP) could be a promising alternative to antibiotics, with most measured parameters in birds fed MLP diets comparing well to those of control and antibiotic-fed broilers. MLP supplementation improved FBW, ADG, and ADFI while reducing FCR, suggesting enhanced digestive physiology and efficient feed conversion for improved growth performance. Additionally, it modulated cecal bacteria, reducing entero-pathogens and increasing beneficial bacteria. The blood analysis revealed higher RBC and HB levels in the MLP-fed broilers, indicating a more oxygenated and healthy blood condition. A lower H/L ratio in the MLP-fed broilers suggests potential for stress management by MLP. Furthermore, lower levels of TG, cholesterol, and abdominal fat indicate the promotion of healthy meat production. Importantly, no adverse health effects were observed in the broilers fed with MLP. While these findings are encouraging, further research is needed to explore the optimal dosage of MLP for broiler production across different breeds and age groups. Additionally, investigating the specific bioactive compounds in MLP responsible for its positive effects would be valuable. Overall, this study paves the way for utilizing MLP as a sustainable and effective feed additive to enhance growth performance, overall well-being, and potentially reduce reliance on antibiotics in the poultry industry.

## Figures and Tables

**Table 1 vetsci-11-00374-t001:** Nutrient composition of experimental basal diet.

Components	Starter (1–21 Days)	Finisher (22–35 Days)
Corn	58.17	64.26
Soybean meal	30.44	24.05
Corn gluteal meal	3.53	4
Soybean oil	3.8	4
DCP ^1,2^	1.5	1.09
Limestone	1.3	1.35
Salt	0.3	0.3
Sodium-bi-Carbonate	0.3	0.3
L-Lysine (98%) ^2^	0.01	0.08
Methionine (98%) ^2^	0.14	0.04
Threonine (98%) ^2^	0.01	0.03
Premix ^3^	0.5	0.5
Total	100	100
**Chemical Analysis**		
Dry Matter (%)	88	87.5
Crude Protein (%)	20.5	19
Crude Fiber (%)	5	5.2
Calcium (%)	1	0.90
Phosphorus (%)	0.45	0.40
**Calculated Value**		
Metabolizable Energy Kcal/kg	2950	3050
Total Phosphorus	0.45	0.45
Digestible lysine	1.15	1.05
Digestible methionine	0.48	0.45

^1^ DCP = Di-Calcium-Phosphate; ^2^ commercially available; ^3^ supplied per kilogram of diet: vitamin A (trans-retinyl acetate), 10,050 IU; vitamin D3, 2800 IU; vitamin E (DL-a-tocopheryl acetate), 50 mg; vitamin K3, 3.5 mg; thiamine, 2.5 mg; riboflavin, 7.5 mg; pantothenic acid,15.3 mg; pyridoxine, 4.3 mg; vitamin B12 (cyanocobalamin), 0.02 mg; niacin, 35 mg; choline chloride, 1000 mg; biotin, 0.20 mg; folic acid, 1.2 mg; Mn, 100 mg; Fe, 85 mg; Zn, 60 mg; Cu, 9.6 mg; I, 0.30 mg; Co, 0.20 mg; and Se, 0.20 mg.

**Table 2 vetsci-11-00374-t002:** Chemical composition of *M. oleifera* leaf powder.

Proximate Components	Value (%)
Dry Matter	91.21
Ash	9.28
Crude Protein (CP)	26.5
Crude Fiber (CF)	7.95
Crude Fat	4.49
Organic matter (OM)	81.93
Fatty acid	2.05
Nitrogen Free Extract (NFE)	42.99
Total Phosphorus (mg) *	442
Calcium (mg) *	71

***** Values are in per 100 g plant leaf powder material.

**Table 3 vetsci-11-00374-t003:** Amino acid composition of *M. oleifera* leaves *****.

Amino Acids	Amount (mg/100 g)
Arginine	1.07
Isoleucine	0.95
Leucine	1.75
Lysine	1.03
Methionine	0.34
Phenylalanine	1.43
Threonine	1.07
Valine	0.89

* These values were analyzed by an L-8900 model Amino Acid Analyzer at the Department of Livestock (DLS) nutrition lab, Farmgate, Dhaka.

**Table 4 vetsci-11-00374-t004:** Effects of *M. oleifera* leaf powder on the growth performance of broilers *.

Items		Control	Antibiotic	MLP	SEM	*p*-Value
Initial body weight (g)		40.23	39.95	39.92	0.16	0.719
Final body weight (g)		2116 ^c^	2142 ^b^	2319 ^a^	22.34	0.001
Avg. Daily Gain (ADG) (g/day/bird)	1–7 day	19.66	19.76	19.99	0.16	0.688
	8–14 day	49.24 ^b^	50.93 ^b^	54.31 ^a^	0.59	0.001
	15–21 day	69.91 ^c^	72.35 ^b^	78.17 ^a^	0.92	0.001
	22–28 day	77.06 ^b^	75.63 ^c^	85.50 ^a^	1.08	0.001
	29–35 day	80.67 ^c^	81.67 ^b^	87.64 ^a^	0.87	0.001
	1–35 day	59.31 ^c^	60.07 ^b^	65.12 ^a^	0.64	0.001
Average Daily Feed Intake (ADFI) (g/day/bird)	1–7 day	20.79	20.68	20.84	0.17	0.931
	8–14 day	53.32 ^c^	54.78 ^b^	57.06 ^a^	0.43	0.001
	15–21 day	103.06 ^c^	106.65 ^b^	115.10 ^a^	1.29	0.001
	22–28 day	160.94 ^a^	142.02 ^c^	145.19 ^b^	2.04	0.001
	29–35 day	190.33 ^c^	193.20 ^b^	201.67 ^a^	1.52	0.001
	1–35 day	105.69 ^b^	103.47 ^c^	107.97 ^a^	0.49	0.001
Feed Conversion Ratio (FCR) (Weekly)	1–7 day	1.058	1.047	1.043	0.01	0.695
	8–14 day	1.084	1.076	1.051	0.01	0.282
	15–21 day	1.475	1.475	1.473	0.01	0.993
	22–28 day	2.089 ^a^	1.878 ^b^	1.698 ^c^	0.04	0.001
	29–35 day	2.361	2.366	2.303	0.02	0.368
	1–35 day	1.782 ^a^	1.723 ^b^	1.658 ^c^	0.01	0.001

* The data represent the mean value of 64 samples per treatment. ^a,b,c^ Values with different superscripts within the same row indicate a significant difference at *p* < 0.05; SEM: pooled standard error of the means. Level of significance at *p* < 0.05; Antibiotic: chlortetracycline (CTC), at 75 mg/kg; MLP: at 100 mg/kg; Abbreviations: MLP = *M. oleifera* leaf powder.

**Table 5 vetsci-11-00374-t005:** Effect of *M. oleifera* leaf powder on carcass characteristics of broilers *.

Traits	Control	Antibiotic	MLP	SEM	*p*-Value
Dressed wt.%	64.13	64.65	65.79	0.50	0.431
Wing wt.%	6.31	6.07	6.15	0.09	0.610
Drum stick wt.%	8.26 ^b^	8.29 ^b^	9.26 ^a^	0.22	0.038
Thigh wt.%	9.42 ^b^	9.55 ^ab^	10.98 ^a^	0.36	0.043
Shank wt.%	3.77	3.59	3.64	0.04	0.184

* The data represent the mean value of eight samples per treatment. ^a,b^ Values with different superscripts within the same row indicate a significant difference at *p* < 0.05; SEM: pooled standard error of the means. Level of significance at *p* < 0.05; antibiotic: chlortetracycline (CTC)—75 mg/kg; MLP: 100 mg/kg; abbreviations: MLP = *M. oleifera* leaf powder; wt. = weight.

**Table 6 vetsci-11-00374-t006:** Effects of *M. oleifera* leaf powder on weight of inner organs of broilers (in percent of live body weight) *.

Inner Organs wt. (%)	Control	Antibiotic	MLP	SEM	*p*-Value
Heart	0.59	0.56	0.56	0.02	0.530
Liver	2.74	2.82	2.43	0.10	0.300
Pancreas	0.34	0.29	0.27	0.01	0.246
Spleen	0.16 ^a^	0.14 ^b^	0.10 ^c^	0.01	0.040
Proventriculus	0.43	0.42	0.45	0.01	0.060
Gizzard	1.47	1.46	1.49	0.03	0.472
Intestine	5.67	5.38	5.93	0.20	0.632
Bursa	0.17	0.13	0.15	0.02	0.549
Abdominal Fat	1.12 ^a^	1.14 ^a^	1.01 ^b^	0.03	0.052

* The data represent the mean value of eight samples per treatment. ^a,b,c^ Values with different superscripts within the same row indicate a significant difference at *p* < 0.05; SEM: pooled standard error of the means. Level of significance at *p* < 0.05; antibiotic: chlortetracycline (CTC), at 75 mg/kg; MLP: 100 mg/kg; abbreviations: MLP = *M. oleifera* leaf powder; wt. = weight.

**Table 7 vetsci-11-00374-t007:** Effects of *M. oleifera* leaf powder on meat quality of broilers *.

Traits		Control	Antibiotic	MLP	SEM	*p*-Value
pH	45 min	6.02 ^b^	6.20 ^a^	6.35 ^a^	0.06	0.010
	24 h.	5.64	5.69	5.76	0.03	0.235
Drip loss (%)	24 h.	2.70	2.64	2.40	0.09	0.446
	7 days	5.12	5.16	4.58	0.226	0.653
Cooking loss (%)		28.47 ^a^	26.90 ^a^	23.31 ^b^	0.81	0.001

* The data represent the mean value of eight samples per treatment. ^a,b^ Values with different superscripts within the same row indicate a significant difference at *p* < 0.05; SEM: pooled standard error of the means. Level of significance at *p* < 0.05; antibiotic: chlortetracycline (CTC), at 75 mg/kg; MLP: 100 mg/kg; abbreviations: MLP = *M. oleifera* leaf powder; h. = hours.

**Table 8 vetsci-11-00374-t008:** Effects of *M. oleifera* leaf powder on biochemical parameters of blood of broilers *.

Parameters	Control	Antibiotic	MLP	SEM	*p*-Value
Blood creatinine (mg/dL)	0.30	0.27	0.23	0.01	0.318
SGPT(ALT) (U/L)	10	7.4	7.8	0.69	0.238
Total Cholesterol (mg/dL)	212 ^a^	276 ^a^	150 ^b^	21.8	0.034
Triglycerides (TG) (mg/dL)	91 ^a^	97 ^a^	71 ^b^	4.95	0.040

* The data represent the mean value of eight samples per treatment. ^a,b^ Values with different superscripts within the same row indicate a significant difference at *p* < 0.05; SEM: pooled standard error of the means. Level of significance at *p* < 0.05; antibiotic: chlortetracycline (CTC), at 75 mg/kg; MLP: 100 mg/kg; abbreviations: MLP = *M. oleifera* leaf powder; SGPT = serum glutamic pyruvic transaminase; ALT = alanine aminotransferase.

**Table 9 vetsci-11-00374-t009:** Effects of *M. oleifera* leaf powder on hematological parameters of broilers *.

Test Parameters		Control	Antibiotic	MLP	SEM	*p*-Value
Hemoglobin (Hb) (gm/dl)		10 ^b^	11 ^b^	13.8 ^a^	0.57	0.041
Red blood cells (RBC) (Million/cmm)		2.3 ^b^	2.56 ^b^	3.09 ^a^	0.08	0.042
Packed cell volume (PCV) %		33	34.8	42.5	1.88	0.179
Mean corpuscular volume (MCV) (fl)		130	134	138	3.44	0.975
Mean corpuscular hemoglobin (MCH) (pg)		40	42	43	2.112	0.771
Mean corpuscular hemoglobin concentration (MCHC) (gm/dL)		31	33	31	1.213	0.741
Red cell distribution width–coefficient of variation (RDW–CV) %		6	7	9	0.162	0.852
White blood cells (WBC) total count (k/cmm)		126	124	123	5.33	0.886
WBC-Differential count (DLC)	Heterophils %	27	26	25	1.73	0.552
	Lymphocyte %	62	62	63	3.22	0.653
	Monocyte %	11	12	12	0.88	0.941
	H/L ^1^	0.43 ^a^	0.41 ^b^	0.39 ^c^	0.03	0.041
Platelet (total count) (k/cmm)		34	33	34	1.53	0.631

* The data represent the mean value of eight samples per treatment. ^a,b,c^ Values with different superscripts within the same row indicate a significant difference at *p* < 0.05; SEM: pooled standard error of the means. Level of significance at *p* < 0.05. ^1^ H/L = heterophils: lymphocyte (Stress indicator); antibiotic: chlortetracycline (CTC), at 75 mg/kg; MLP: 100 mg/kg; abbreviations: MLP = *M. oleifera* leaf powder; cmm = cells per cubic millimeter; k/cmm = cells per microliter (µL); fl = femtoliters; pg = picogram.

**Table 10 vetsci-11-00374-t010:** Effects of *M. oleifera* leaf powder on cecal bacteria of broiler *.

Microbial Species	Control (log10cfu/gm)	Antibiotic (log10cfu/gm)	MLP (log10cfu/gm)	SEM	*p*-Value
*Escherichia coli*	6.19 ^a^	6.07 ^b^	6.08 ^b^	0.02	0.003
*Klebsiella* spp.	5.65	5.63	5.64	0.02	0.961
*Salmonella* spp.	5.87	5.82	5.83	0.02	0.071
*Shigella* spp.	5.96 ^a^	5.90 ^b^	5.87 ^c^	0.01	0.001
*Lactobacillus* spp.	6.21 ^b^	6.18 ^b^	6.37 ^a^	0.03	0.001
Total aerobic plate count (TAPC)	7.41 ^a^	7.33 ^b^	7.43 ^a^	0.01	0.006

* The data represent the mean value of eight samples per treatment. ^a,b,c^ Values with different superscripts within the same row indicate a significant difference at *p* < 0.05; SEM: pooled standard error of the means. Level of significance at *p* < 0.05; antibiotic: chlortetracycline (CTC), at 75 mg/kg; MLP: at 100 mg/kg; abbreviations: MLP = *M. oleifera* leaf powder; CFU = colony forming unit.

## Data Availability

The data are available within this article.

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
