# Peer review of "Effects of Moringa oleifera Leaf Powder on the Growth Performance, Meat Quality, Blood Parameters, and Cecal Bacteria of Broilers"

_vetsci, 2024, doi:10.3390/vetsci11080374_

Round 1

Reviewer 1 Report

Comments and Suggestions for Authors

An interesting approach to replacing the use of antibiotics. I hope they continue the line of work.

Author Response

Reviewer -1:

An interesting approach to replacing the use of antibiotics. I hope they continue the line of work.

Authors Responses: Dear Reviewer, thank you very much for reviewing our manuscript with good necessary comments, that’s all have been highly accepted by us. Thanks a lot for your valuable time for us.

Reviewer 2 Report

Comments and Suggestions for Authors

Dear authors,

in your manuscript you presented the results of the investigation related to the use of Moringa oleifera leaf powder as growth promotor in broilers, and its effect on growth performance, meat quality and cecal microbiota.

In the whole text please uniform writing of Moringa oleifera (Italic, capital letters, etc) and use of MOL, MLP, MOLM, etc.

Please include into title "blood parameters"; in the title you mention "cecal microbiota" and later in the text you showed the results of  specific bacteria  as E. coli, Salmonella, Shigella, Klebsiella and Lactobacillus?

L 48 please rephrase "supplement for growth promoter"

Chapter 2.2 - please include the details about weighting of the animals, if done

L 158 please check this sentence (verb missing?)

L 162 dot missing at the end of the sentence.

L 227 please mention the manufacturer(s)

L 232 and 237 please write bacteria with capital letter

Chapter 2.8. please describe how did you detect "total aerobic plate count" as mentioned in L 351; please mention the manufacturer(s) of the agars

L 253 please rephrase "the role of feeding Moringa"

L 263-268 this paragraph needs revision

L 294 please rephrase "on inner organs" - on what of inner organs - weight, size, percentage?

L 318-319 please check this sentence (begins with "while")

Table 8 - no superscripts here

L 347,348, 350 E. bacteria italic?

L 360-363 sentence needs revision (potential effect; only mentioning broiler production - what about the effect on other examined parameters)?

L 490-494 sentences needs revision

I found almost similar research, published couple of years ago, titled: Effect of Dietary Moringa oleifera Leaves Powder on Growth Performance, Blood Chemistry, Meat Quality and Gut Microflora of Broiler Chicks (authors Divya, A.B. Mandal, A. Biswas, A.S. Yadav1 and A.K. Biswas), published in Animal Nutrition and Feed Technology (2014) 14: 349-357. Authors of received manuscript didn't cited this paper from 2014.  I recommend reconsider after major revision.

Author Response

Dear authors,

in your manuscript you presented the results of the investigation related to the use of Moringa oleifera leaf powder as growth promotor in broilers, and its effect on growth performance, meat quality and cecal microbiota.

Authors responses: Dear Reviewer, thank you very much for reviewing our manuscript and providing valuable comments. We have carefully considered all your suggestions and revised the manuscript accordingly. All changes have been highlighted in yellow in the text.

Reviewer comment 1: In the whole text please uniform writing of Moringa oleifera (Italic, capital letters, etc) and use of MOL, MLP, MOLM, etc.

Authors responses: We have carefully considered your suggestions and revised the manuscript accordingly. All changes have been highlighted in yellow in the text. Thank you.

Reviewer comment 2: Please include into title "blood parameters"; in the title you mention "cecal microbiota" and later in the text you showed the results of specific bacteria as E. coli, Salmonella, Shigella, Klebsiella and Lactobacillus?

Authors responses: The title is changed accordingly. And we have modified in text as well. Please check the L31,235,362,370 and 518. Thank you. Please check the title as below

Effects of Moringa oleifera Leaf Powder on the Growth Performance, Meat Quality, Blood Parameters and Cecal Bacteria of Broilers

In the title, we have added “blood parameters” and modified “cecal microbiota” to “cecal bacteria” as we focused on four pathogenic bacteria and one beneficial bacterium as representatives of the major cecal bacteria. This approach allowed us to assess how Moringa affects the numbers of beneficial and pathogenic bacteria in the cecum. We also determined the Total Aerobic Plate Count (TAPC) to provide an overview of the total population of major cecal bacteria. Thank you.

Reviewer comment 3: L 48 please rephrase "supplement for growth promoter"

Authors responses: Rephrased. Please check the L50. Thank you.

Reviewer comments 4: Chapter 2.2 - please include the details about weighting of the animals, if done

Authors responses: We have provided the necessary information regarding the weighting of birds, average daily weight gain, and average daily feed intake in Section 2.4. Please check this section. Please check the L179-181. Thank you.

Reviewer comments 5: L 158 please check this sentence (verb missing?)

Authors responses: Verb is added. Please check the L161. Thank you.

Reviewer comments 6: L 162 dot missing at the end of the sentence.

Authors responses: Full-stop is added. Please check the L165. Thank you.

Reviewer comments 7: L 227 please mention the manufacturer(s)

Authors responses: Manufacturer’s names are mentioned. Please check the L228 and 233. Thank you.

Reviewer comments 8: L 232 and 237 please write bacteria with capital letter

Authors responses: Corrected accordingly. Please check the L239,251 and 252. Thank you.

Reviewer comments 9: Chapter 2.8. please describe how did you detect "total aerobic plate count" as mentioned in L 351; please mention the manufacturer(s) of the agars.

Authors responses: Described accordingly. And also added the manufacturer’s name of the agars. All of the agars were obtained from “Hi-Media, Mumbai, India”.  Please check the L254-256. Thank you.

Reviewer comments 10: L 253 please rephrase "the role of feeding Moringa"

Authors responses: We’ve rephrased as per suggestion. Please check the L272. Thank you.

Reviewer comments 11: L 263-268 this paragraph needs revision

Authors responses: We’ve revised accordingly. Please check the L282-285. Thank you.

Reviewer comments 12: L 294 please rephrase "on inner organs" - on what of inner organs - weight, size, percentage?

Authors responses: Corrected. Please check the L310. Thank you.

Reviewer comments 13: L 318-319 please check this sentence (begins with "while")

Authors responses: Revised. Please check the L334-336. Thank you.

Reviewer comments 14: Table 8 - no superscripts here.

Authors responses: Corrected. Please check the Table 8. Thank you.

Reviewer comments 15: L 347,348, 350 E. bacteria italic?

Authors responses: Corrected accordingly. Please check the L363,365 and 367. Thank you.

Reviewer comments 16: L 360-363 sentence needs revision (potential effect; only mentioning broiler production - what about the effect on other examined parameters)?

Authors responses: Revised accordingly. Please check the L377-379. Thank you.

Reviewer comments 17: L 490-494 sentences need revision

Authors responses: Revised accordingly. Please check the L507-511. Thank you.

Hope you will consider our revised submission and provide us with the opportunity to share our research findings with the scientific community.

Reviewer 3 Report

Comments and Suggestions for Authors

This article summarizes the uses, values, functions and effects of moringa leaf powder on gut health. This study investigated the effects of low concentrations of MLP on overall growth performance, carcass characteristics, meat quality, blood biochemical characteristics, and gut microbiota of broilers impact. The author is correct about the development of this field, but he does not have his own novel and unique academic views on the future development trend. There are several aspects to note in this article. I recommend this for publication after the authors have addressed the following.

1. In paragraph 28 it is written in 6 repetitions of 6 chickens, while in paragraph 118 it is written in 8 repetitions of 8 chickens?

2. Why only chlortetracycline was used in the antibiotic group?

3. How to determine the dosage of MLP is there a literature reference?

4. Line147, Moringa oleifera leaves were collected from different periods, but the experimental group did not clearly state which period, and is there a difference between different periods?

5. Line215, Why was there only one chicken per group?

6. Why is there a significant difference in the weight of chicken thighs in the MLP group?

7. Are there any experimental proofs of anti-inflammatory properties in the article?

Author Response

This article summarizes the uses, values, functions and effects of moringa leaf powder on gut health. This study investigated the effects of low concentrations of MLP on overall growth performance, carcass characteristics, meat quality, blood biochemical characteristics, and gut microbiota of broilers impact. The author is correct about the development of this field, but he does not have his own novel and unique academic views on the future development trend. There are several aspects to note in this article. I recommend this for publication after the authors have addressed the following.

Authors responses:  Dear Reviewer, thank you very much for reviewing our manuscript and providing valuable comments. We have carefully considered all your suggestions and revised the manuscript accordingly. All changes have been highlighted in yellow in the text. Below the point-to- point responses for your kind consideration.

Reviewer comment 1: In paragraph 28 it is written in 6 repetitions of 6 chickens, while in paragraph 118 it is written in 8 repetitions of 8 chickens?

Authors responses: Corrected. It was a typo mistake. We use 8 replications of 8 chickens in our experiment. Please check the L36. Thank you.

Reviewer comment 2:  Why only chlortetracycline was used in the antibiotic group?

Authors responses: In our country, the Chlortetracycline group of antibiotics is commonly and randomly used as a growth promoter in poultry production. That is why we used it as a representative group of antibiotic growth promoters. Thank you.

Reviewer comment 3: How to determine the dosage of MLP is there a literature reference?

Authors responses: In the literature, we found that a small dose is much more efficient in broilers than in laying hens and ruminants.  M. oleifera leaves contain some anti-nutritional factors, such as tannins and saponins, in higher doses, which may decrease the digestibility and palatability of the ration. We hypothesize that a higher dosage of Moringa may create a pungent odor and taste in the diets, which could affect feed intake in broilers. Additionally, dietary supplementation with Moringa leaves may add extra fiber to the diets, potentially leading to poor digestion in broilers. Considering these issues, we used a minimal dose of M. oleifera in this broiler experiment. Hope you will consider the fact. Thank you.

Reviewer comment 4: Line147, Moringa oleifera leaves were collected from different periods, but the experimental group did not clearly state which period, and is there a difference between different periods?

Authors responses: Sorry for the misunderstanding. We collected leaves at different mature stages and mixed them uniformly. We have revised the necessary information in the text as suggested. Please check the L150. Thank you.

Reviewer comment 5: Line215, Why was there only one chicken per group?

Authors responses: We randomly choose one chicken from each replication and a total of 8 chicken per group. Please check the L221. Thank you.

Reviewer comment 6: Why is there a significant difference in the weight of chicken thighs in the MLP group?

Authors responses: A higher thigh weight may be corelated with the higher final body weight of broilers in the MLP group. Thank you.

Reviewer comment 7. Are there any experimental proofs of anti-inflammatory properties in the article?

Authors responses: Sorry, we did not analyze additional anti-inflammatory parameters such as inflammatory cytokines (IL-2, IL-4, IL-10, TNF-α, IFN-γ, etc.). However, we have analyzed kidney function (blood creatinine) and liver function tests (SGPT). We will consider your suggestions in our future study. Thank you.

Hope you will consider our revised submission and give us the opportunity to contribute our research findings to the scientific community.

Round 2

Reviewer 2 Report

Comments and Suggestions for Authors

Dear authors,

thank you for taking into consideration the suggestion given by the reviewer.

Only, you didn't comment this part of the review:

I found almost similar research, published couple of years ago, titled: Effect of Dietary Moringa oleifera Leaves Powder on Growth Performance, Blood Chemistry, Meat Quality and Gut Microflora of Broiler Chicks (authors Divya, A.B. Mandal, A. Biswas, A.S. Yadav and A.K. Biswas), published in Animal Nutrition and Feed Technology (2014) 14: 349-357. 

If investigations are comparable, please do so and include this reference in your text. 

Author Response

Reviewer 2:

Comment 1: Dear authors,

thank you for taking into consideration the suggestion given by the reviewer.

Author response: Dear Reviewer, You’re welcome.

Thank you very much for reviewing our manuscript and providing valuable comments. We greatly appreciate your time and effort.

Comment 2: Only, you didn't comment this part of the review:

I found almost similar research published couple of years ago, titled: Effect of Dietary Moringa oleifera Leaves Powder on Growth Performance, Blood Chemistry, Meat Quality and Gut Microflora of Broiler Chicks (authors Divya, A.B. Mandal, A. Biswas, A.S. Yadav and A.K. Biswas), published in Animal Nutrition and Feed Technology (2014) 14: 349-357. 

If investigations are comparable, please do so and include this reference in your text. 

Author response: Thank you for pointing this out. We sincerely apologize for the oversight. Your valuable suggestion is greatly appreciated, as the mentioned article is relevant and comparable to our research work. Please check L502-504 and reference number 65. Thank you.

Hope you will consider our revised submission and give us the opportunity to contribute our research findings to the scientific community.

Reviewer 3 Report

Comments and Suggestions for Authors

Thank you.

Author Response

Comment: Thank you.

Author response: Dear Reviewer, You’re welcome.
Thank you very much for reviewing our manuscript and providing valuable comments. We greatly appreciate your time and effort.

Round 3

Reviewer 2 Report

Comments and Suggestions for Authors

Dear authors

Thank you for your reply.